# Single-Cell RNA-Seq Reveals Conserved Cellular Communication Mechanisms Governing Ocular Lineage Specification from Human iPS Cells

**DOI:** 10.3390/cells15020104

**Published:** 2026-01-07

**Authors:** Laura Howard, Yuki Ishikawa, Rei Kamuro, Tomohiko Katayama, Kiranjit K. Bains, Matthew J. Hill, Derek J. Blake, Sung-Joon Park, Ryuhei Hayashi, Andrew J. Quantock, Kohji Nishida

**Affiliations:** 1School of Optometry and Vision Sciences, Cardiff University, Cardiff CF24 4HQ, Wales, UK; 2Centre for Neuropsychiatric Genetics and Genomics, School of Medicine, Cardiff University, Cardiff CF24 4HQ, Wales, UK; 3Department of Stem Cells and Applied Medicine, Osaka University Graduate School of Medicine, Osaka 565-0871, Japan; 4Department of Ophthalmology, Osaka University Graduate School of Medicine, Osaka 565-0871, Japan; 5Institute for Open and Transdisciplinary Research Initiatives, Osaka University, Osaka 565-0871, Japan; 6Laboratory of AI Genome Informatics, Department of Frontier Research and Development, Kazusa DNA Research Institute, Chiba 292-0818, Japan

**Keywords:** SEAM, hiPSC, ocular, single-cell, transcriptomics, signalling

## Abstract

The complexity of cell fate decisions that underpin early eye development can be effectively modelled by leveraging the unique properties of human induced pluripotent stem cells (hiPSCs). In this study, we have utilised transcriptomic data generated from hiPSCs as they begin to self-organise and differentiate into two-dimensional eye-like organoids in vitro, and employ advanced single-cell analytical tools to dissect the cellular communication networks that direct this dynamic process. We have identified key signalling mediators and transcriptional effectors that guide the transition from pluripotency through to ocular differentiation, and our analyses reveal the conservation of developmentally defined signalling pathways. Members of the Activin, FGF, BMP, WNT, and retinoic acid families of ligands and receptors displayed communication probabilities consistent with their ocular-specific developmental roles in vivo, and this was accompanied by conserved tissue-specific activity of transcriptional regulators. These findings not only highlight the utility of hiPSCs for studying the cellular interactions and molecular pathways that drive early developmental decisions, but also advance our understanding of eye development in an accessible stem cell-based system.

## 1. Introduction

Eye development is a complex and tightly regulated process which involves intricately coordinated interactions between neuroectoderm, surface ectoderm, and neural crest-derived periocular mesenchyme. Communication between interacting cells is essential for multicellularity, and in the developing eye, underpins how cells with distinct tissue origins structurally and functionally coordinate to generate a sensory organ capable of sight. For example, while the corneal epithelium derives from embryonic surface ectoderm, the highly specialised cells of the retina develop from neuroectoderm, and it is the interaction of these tissues with one another and with their extracellular environment that drives stereotyped three-dimensional morphogenesis. A key challenge, therefore, is to understand the genetic principles that underlie the cell fate and lineage commitment decisions that direct divergent pathways, and to ascertain how stem and progenitor cells are coordinated to drive cellular differentiation, tissue organisation, and organ formation during development.

Human induced pluripotent stem cells (hiPSCs), which self-renew indefinitely and can be differentiated into virtually any of the specialised cell types found in the human body, are ideal for studying early cell interactions in vitro. hiPSCs can be induced in culture to generate a self-formed, ectodermal, autonomous multi-zone (SEAM) [1], a growing two-dimensional primordium of ocular cells which progressively self-organises into concentric zones that are indicative of lineage, recapturing aspects of whole-eye morphogenesis. A mature SEAM consists of four readily identifiable zones, which represent neuronal cells (Zone 1), retinal pigment epithelium and neural retina (Zone 2), ocular epithelium (Zone 3), and non-ocular epithelium (Zone 4) [1,2]. The generation of ocular tissue from hiPSCs offers huge potential in regenerative medicine, and SEAM-derived epithelial cells have previously been isolated and shown to recover function when transplanted onto experimental ocular wounds [1,3]. Corneal epithelial cell sheets derived from SEAMs have now also been successfully transplanted in-human, with encouraging clinical outcomes [4], advancing therapeutic application of this hiPSC technology. We recently reported results obtained from a large-scale single-cell transcriptomic interrogation of developing SEAMs over a 12-week period, which allowed us to identify the multiple specialised ocular cell types that are generated over the course of SEAM maturation [5]. Here, we utilise and extend our transcriptomic analysis to focus on and explore the initial phases of SEAM development when hiPSCs transition away from pluripotency and begin to commit to ectodermal cell fates. By defining the major signalling pathways and communication patterns that influence these early cell fate decisions, our study explores both the key principles underlying cell fate control in hiPSCs and the orchestration of the decisions that shape the formation of the developing human eye.

## 2. Materials and Methods

### 2.1. SEAM Cell Culture

SEAM cell culture was performed as described previously [1,2,6]. Briefly, hiPSCs (clone 201B7 [7], RIKEN BioResource Center, Tsukuba, Japan) were cultured on 0.5 μg/cm^2^ LN511E8-coated dishes (iMatrix-511 silk, 892-021, Nippi, Tokyo, Japan) in serum-free StemFit medium (AK03N, Ajinomoto, Tokyo, Japan) for at least two passages to ensure stability. Cells were harvested using dissociation solution (DS) containing 50% TrypLE Select (12563-029, Thermo Fisher Scientific, Waltham, MA, USA) and 50% 0.5 mmol/L EDTA/PBS (13567-84, Nacalai Tesque, Kyoto, Japan) and seeded at 4500 cells per well in 6-well plates (353-046, Corning Incorporated, Corning, NY, USA) coated with 0.5 μg/cm^2^ LN511E8. Cells were then cultured for a further 10 days in StemFit medium, and differentiation was initiated by culture in differentiation medium (DM) following established protocols [2]. Medium changes were performed once every 2–3 days throughout the SEAM cell culture period using the media compositions defined in Appendix A. Whole SEAMs were collected from each well at successive time points spanning early differentiation: using DS for 4 min on day 10, TrypLE Express (12604-013, Thermo Fisher Scientific) for 10 min after +1 week, or Accutase (12679-54, Nacalai Tesque) for 10 min after +2 weeks. Single-cell suspensions containing cells from all zones were isolated and sorted on a FACSAriaII cytometer (BD Biosciences, Franklin Lakes, NJ, USA) using 7-AAD (559-925, BD Biosciences) staining to identify viable cells. Cells were processed according to the latest available 10x Genomics protocols, and single-cell libraries were prepared using the Chromium Next GEM Single Cell 3′ Reagent Kits v3.1 (PN-1000127, 10x Genomics, Pleasanton, CA, USA) according to the manufacturer’s instructions. Encapsulation of cells into GEMs was performed using a Chromium Single Cell Controller with targeted recovery of 5000 cells per time point. Libraries were constructed following standard user guidelines and sequenced on an Illumina NovaSeq 6000 platform using a high-output flow cell with 10x Genomics dual indexing to generate paired-end reads.

### 2.2. Single-Cell Data Processing and Analysis

Raw sequencing data files were transformed into single-cell gene count matrices using Cell Ranger 6.1.1 and mRNA reads were aligned to the human reference genome GRCh38-2020-A. Data pre-processing and cell cluster analyses were performed in R using Seurat v4 [8,9]. Initially, quality control filters were applied to the raw data to filter out cells with unique feature count values less than 500 or with greater than 15% mitochondrial reads. Following pre-processing, additional putative low-quality cells were removed, and doublets were excluded with DoubletFinder [10]. Normalisation and scaling were performed using SCTransform v2 [11] and dimensionality reduction was performed by PCA and UMAP embedding. Clustree [12] was used to assess cluster stability and a clustering resolution of 0.8 was used in all downstream analyses. Differential expression analysis was performed using the FindAllMarkers function with min.pct = 0.25 and logfc.threshold = 0.25. Monocle3 [13,14] was used for trajectory analysis. Seurat objects were converted into Monocle cell data sets using the as.cell_data_set function from SeuratWrappers and the learn_graph() function was applied to fit a principal graph. The root of the trajectory was determined programmatically and order_cells() was applied to order the cells in pseudotime. To analyse genes that change as a function of pseudotime, graph_test() was passed to neighbor_graph = “principal_graph” and find_gene_modules() applied with resolution = 1 × 10^−4^. Trajectory-dependent genes were assessed in the context of our previously published full 12-week SEAM developmental trajectory [5] by setting clear_cds = F when running choose_graph_segments. Cellular potency scores were determined using CytoTRACE2 [15]. For CellChat analysis [16,17], Seurat objects were transformed into CellChat objects, and the ligand–receptor database was set to CellChatDB.human. Communication probabilities were quantified via mass action models followed by permutation testing to determine significant interactions, using in-built CellChat functions. Visualisations and systems analyses of communication networks were performed using recommended parameters. Relative activity scores for key pathways are shown in Appendix A. For SCENIC analysis [18], individual Seurat objects from the 3 timepoints were integrated using the merge function, and PrepSCTFindMarkers was applied before differential expression analysis to correct for heterogeneity in sequencing depth. The combined data file was converted into a loom file, and GENIE3 was used to infer co-expression networks. RcisTarget was used for transcription factor (TF) motif enrichment analysis with databases set to hg38_500bp_up_and_100bp_down_tss and hg38_10kb_up_and_down. AUCell was used to identify cells with active gene sets and calculate regulon activity scores using default settings throughout. RegulonAUC values were passed to calcRSS and plotted to generate regulon specificity score (RSS) dotplots. Unless specified, only high-confidence annotations are presented. Regulons are reported using nomenclature TF_(ng) to identify the TF driver and the number (n) of gene targets (g) contained in the TF regulon. Metascape analysis was performed at metascape.org using default input parameters, with gene lists derived from SCENIC regulons.

## 3. Results

### 3.1. Divergence of Neuroectodermal and Surface Ectodermal Trajectories in Early SEAM Development

Two principal pathways representing divergent cellular lineages become evident during the earliest stages of SEAM formation [5]. Their inferred trajectories chart the transition from broad hiPSC pluripotency into either neuroectoderm (NE) (which will give rise to the neural and retinal cells of Zones 1 and 2) or into surface ectoderm (SE) (which will give rise to the epithelial cells of Zones 3 and 4). We began our current study by examining a subset of data [5] obtained from cells harvested over the first three weeks of SEAM culture (Appendix A). By this stage, immature SEAMs have begun to loosely self-organise into a characteristic concentric multizone (Figure 1A), although the zonal boundaries shown schematically in Figure 1B are yet to be sharply defined. To investigate changes in trajectory-dependent genes that might initiate differentiation along divergent routes, we used Monocle3 [13,14] to analyse differential expression in just these early cells whilst preserving the known trajectory of SEAM development [5] through to maturity (Figure 1C,D). Trajectory-variable gene expression values were extracted using Moran’s I test for spatial autocorrelation [13] and plotted according to inferred time (pseudotime) independently along each of the pathways (Figure 1E). Cells positioned close to the root of the trajectory robustly expressed several key pluripotency markers, including *POU5F1*, *DPPA4* and *TDGF1*. Over time, cells began to differentiate towards either specific SE or NE lineages, which were each characterised by distinct indicators reflective of their respective cellular fates. For instance, SE cells scored highly for the keratin family members *KRT8*, *KRT18*, and *KRT19*, while NE cells were typified by elevated *PAX6*, *SIX6*, and *NEUROD1* expression (Figure 1E). However, we also detected transient, trajectory-dependent upregulation of transcripts encoding multiple signalling mediators and interactors over developmental pseudotime. For example, in SE, the WNT modulators *FRZB* and *TPBG*, and eye-field transcription factor *SIX3*, which is known to suppress WNT signalling and regulate *PAX6* to promote ocular differentiation [19], were transiently upregulated, together with *GPC3*, which encodes a cell surface proteoglycan known to regulate multiple signalling pathways, including those activated by SHHs, WNTs, BMPs, and FGFs [20]. In NE, pseudotime analyses projected transient upregulation of *ASPM*, a multi-pathway signalling modulator which is required for symmetric divisions of proliferative neuroepithelial cells [21], *PRC1*, which acts as an epigenetic repressor of retinoic acid (RA)-responsive genes [22,23], and the NOTCH non-canonical ligand family member *DLK1* [24] (Figure 1E). This prompted us to further explore the influence of cellular communication via signalling ligands, receptors, and cofactors during the earliest phases of SEAM development, when cells begin to move away from a pluripotent state.

### 3.2. Cellular Communication Networks Guiding the Transition from Pluripotency

Cellular differentiation is influenced by highly regulated pathways of cell–cell signalling which cooperate to activate transcriptional programmes and ensure the reproducibility of developmental decisions. After 10 days of culture in pluripotency maintenance medium, differential expression analysis revealed already-marked differences in the transcriptomic profiles of cells according to their differentiation status (Figure 2A and Appendix A). To investigate which signalling networks operate during the very earliest stages of hiPSC/SEAM establishment to drive this, we probed our 10-day datapoint using the R package CellChat v2 [16,17], which integrates input single-cell expression data with curated signal databases to quantitatively model probabilities of intercellular communication. We compared expression levels of pluripotency markers extracted from our Monocle3 analysis (Figure 1E) and the eye-field transcription factor *PAX6* in cells that were grouped by Seurat cluster (Figure 2B and Appendix A), confirming that expression of *PAX6* was higher in cells expressing lower levels of these markers of stemness. Communication probabilities were calculated on a signalling pathway level of all ligand and receptor interactions, and an aggregated cell–cell network inferred extensive communication between all clusters in the dataset (Appendix A). We next investigated the contribution of global communication patterns from within CellChat to explore how cell groups and signalling pathways coordinate during early differentiation. Cells belonging to the clusters depicted in Figure 2B were segregated into groups based on their dominant cell communication patterns (Figure 2C), and influential signalling pathway contributions of outgoing (secreting) cells and incoming (target) cells were identified. Notably, the more differentiated cells in this dataset, belonging to clusters 1, 8, 9, and 10, were characterised by outgoing pattern 2. This pattern represents multiple signalling pathways, including but not limited to NGL, NRG, ncWNT, RA, GABA-B, PTN, and FLRT. Meanwhile, cells belonging to clusters 6 and 7 were represented by pattern 3 and the pluripotency-associated SPP1, L1CAM, and Activin pathways [25,26,27] (Figure 2C), and cells in these clusters expressed the highest levels of *POU5F1* (Figure 2B).

In total, 68 active signalling networks and their components were detected at the earliest stages of hiPSC culture, even before the cells were exposed to differentiation medium to promote SEAM formation. These signalling patterns provide insight into how cells and signals cooperate during early development. FLRT signalling, for example, was strongly associated with the differentiating cells in clusters 1, 8, 9, and 10 (Figure 2C), and FLRTs have been reported to modulate cell adhesion and potentiate FGF signalling [28]. In turn, FGFs regulate diverse biological processes spanning cell proliferation, migration, and differentiation, and play combinatorial roles during eye development [29]. In our data, FGF2 signalling via FGFR1 was most strongly detected from source cells in clusters 2–7 and 11–12, while FGF8 was secreted by cells in clusters 9 and 10 (Appendix A), reflecting their respective roles in promoting pluripotency and neural differentiation [30,31], and these signals might be amplified by FLRTs or other co-modulators. Activin/Nodal signalling is also thought to cooperate with FGF activity to maintain cellular pluripotency and promote the self-renewal of embryonic stem cells [32,33], and we found that in our 10-day-old hiPSC/SEAMs, Activin signalling networks originated from cells in the highly pluripotent cluster 6 only. In contrast, outgoing RA and ncWNT signalling was restricted to cells of cluster 10, and there was multi-cluster engagement with signal receivers and influencers (Figure 2D), whereas FGF signalling pathway networks were more decentralised, with dominant signalling roles shared by senders, receivers, mediators, and influencers.

The bioactivities of various components of these signalling pathways operate under stringent control measures, which help determine the strength, range, and duration of a given signal during development. To explore this further, we analysed the expression of several signalling antagonists relevant to the BMP, TGFβ, Activin, Nodal, and WNT pathways. Of these, we found that *follistatin* (*FST*), a secreted Activin and BMP antagonist, was robustly expressed in most cells. However, expression was notably weaker in cells in clusters 6 and 7, and these cells expressed higher levels of pluripotency-associated *NANOG* (Figure 2E). Moreover, there was a clearly defined gradient of *FST* expression illustrated by UMAP overlay, which increased to reach the highest levels in more differentiated cells (Figure 2F), suggesting that blockade of Activin and/or BMP signalling might promote early differentiation of hiPSCs into immature SEAMs.

### 3.3. Cellular Communication Networks Guiding Lineage Specification in Early SEAMs

We extended our CellChat analysis to differentiating hiPSCs that had been cultured for a further week (+1 WK), but now in differentiation medium to promote SEAM formation (Figure 1A and Appendix A). Cells were again grouped by Seurat cluster, and differential expression analysis revealed widespread expression of the early ectodermal lineage marker *OTX2* in cells belonging to all clusters (Figure 3A). However, as reported previously [5], there was evidence of early lineage divergence, with cells belonging to clusters 6, 7, and 11 expressing markers reflective of surface ectodermal and epithelial fates, such as *EPCAM* and *CDH1*. Meanwhile, expression of *POU5F1* in cluster 11 indicated some sustained pluripotency. Cells in the remaining clusters robustly expressed *LHX2* (Figure 3A), which is a marker of neuroectodermal differentiation and, together with *OTX2*, is required for the formation of the optic vesicle during early eye development [34,35]. This divergence of lineage potential was clearly illustrated by UMAP embedding, with 4162 cells belonging to the neuroectodermal (NE) subgroup, and 742 cells to the surface ectodermal (SE) and pluripotent (PP) subgroups (Figure 3B). We annotated cells in the +1 WK dataset according to this new subgrouping, and performed CellChat analysis to explore how these cell groups signal to drive their respective differentiation. Communication patterns in the SE cells were strongly weighted towards networks involving BMP, WNT, HSPG, CLDN, and CDH1, alongside SEMA3, VISFATIN, and PSAP, which have similarly been reported as active pathways driving SE differentiation in ESC-derived SEAMs [36]. Meanwhile, communication patterns in NE cells were dominated by networks involving NCAM, PTN, GDF, FLRT, NRXN, NOTCH, and ncWNT (Figure 3C).

We have previously proposed that the specification of ocular surface ectoderm in hiPSC/SEAMs likely relies on both balanced inhibition of canonical WNT signalling and exposure of cells to BMP4 [37]. To investigate whether this could also be inferred from single-cell data, we separately annotated our +1 WK SE cells as either presumptive ocular SE (cluster 7) or non-ocular SE (cluster 6) and analysed signalling pathway communication probabilities between these groups of cells. Canonical WNT signalling emanating from non-ocular SE (nOSE) was most strongly detected within-cluster, although signalling was also evident towards developing OSE, whereas ncWNT signalling dominated from NE cells (Figure 3D). BMP signalling was notable to/from cells of the nOSE and OSE, and outgoing RA signalling was OSE-exclusive (Figure 3D). We next examined the expression of *SFRP2*, which encodes a negative regulator of canonical WNT signalling. *SFRP2* and *PAX6* were elevated in cells of the OSE compared with those of the nOSE, whereas *SFRP1* expression levels were comparable across the SE-derived cells (Appendix A), supporting a role for SFRP2-mediated WNT inhibition in the specification of hiPSC-derived OSE. Meanwhile, *BMP4* was expressed at similar levels in OSE and nOSE, but was absent in developing NE (Appendix A). We also utilised our communication pattern data to inspect significant interactions between specific ligand and receptor pairs. For instance, our data show that for WNT signalling, communication probabilities are cell-type and cluster-dependent for both canonical and non-canonical modalities, with cells in the OSE cluster signalling via outgoing non-canonical WNT5A compared with canonical WNT6, which marked nOSE. In contrast, NE cells were a source of WNT2B and WNT5B and signalled via their cognate receptors in specific cells (Figure 3E). BMP signalling was predominantly via BMP4 secretion from the early SE cells via BMPR1A + ACVR2B engagement (Appendix A). Divergent FGF family signalling was also noted, with FGF2-FGFR1 signalling to and from pluripotent cells in cluster 11 in contrast to signalling via FGF8 and FGF9 in early OSE and neuroectoderm (Appendix A), and there was pronounced homogenic FLRT3 signalling in these cells (Appendix A).

We also observed significant lineage-dependent variation in components of extracellular matrix (ECM)–receptor networks in our data. For example, dominant collagen and laminin signalling networks varied according to cell lineage and cluster identity, and there were differences in the ECM–receptor communication probabilities between annotated clusters (Appendix A). It was particularly notable that robust laminin signalling via LAMA1/5, LAMB1/2, and LAMC1 was projected (Appendix A), which is likely reflective of the distinct LN511E8 heterotrimeric isoform on which our hiPSC/SEAMs are cultured [38,39]. Additionally, we observed increased heparin sulphate proteoglycan (HSPG) signalling in surface ectodermal vs. neuroectodermal cells (Appendix A). In the developing eye, HSPGs regulate the activity of several key signalling molecules, including FGFs, WNTs, BMPs, laminins, and collagens [40], and it is likely that similar gradients are active during the generation of multi-lineage ocular SEAMs from hiPSCs.

### 3.4. Cell-Type Specific Signalling Pathways Associated with Zone Identity in Developing SEAMs

After a further week of culture in differentiation medium (+2 WK), the concentric multi-zone arrangement of growing SEAMs becomes more defined (Figure 1A). Furthermore, differential expression analysis facilitates the identification of increasingly distinct cellular subtypes, allowing us to reassign cells grouped by Seurat cluster into six categories based upon their expression profiles: neural progenitor cells (NPCs) (Zone 1), lens and retinal progenitors (Zone 2), OSE (Zone 3), nOSE (Zone 4), plus proliferating epithelium (pE), and residual pluripotent cells (Figure 4A). Pluripotent cells remained closely associated with SE cells in the projected UMAP space, with the former expressing *POU5F1* and the latter expressing prototypical epithelial markers, including *CDH1* and *EPCAM* (Figure 4A,B). Presumptive ocular epithelial cells expressed *PAX6* and were located in Seurat cluster 7 (Figure 4B). As described previously, at this stage in SEAM development, we typically observe a mixed pool of lens and retinal progenitors in transcriptomic clustering analyses [5]. These clusters harbour cells that express *VSX2* and *MITF*, which later mark the neural retina and retinal pigment epithelium, respectively, but are co-expressed in the early optic vesicle [41,42,43], and cells expressing *FOXE3*, which is lens-specific [44,45]. The remaining cells were assigned an NPC identity and expressed well-characterised markers, including *NES*, *SOX2*, and *DCX* (Figure 4B).

We explored the global communication patterns that exist between these defined cell types in the early SEAM, plotted the contribution score for each enriched signalling pathway in the secreting and target cells (Figure 4C,D), and found that cellular identity was closely associated with pathway communication contribution scores. We initially focused on signalling from the lens and RPC subgroup and found a strong contribution from several pathways, including RA, RELN, VTN, TGFβ, THBS, and TULP. Interestingly, BMP signalling was also correlated with the lens and RPCs at this stage, likely reflecting the role of BMP signalling in orchestrating lens induction in vivo [46,47,48], and BMP association was also directly correlated with ocular epithelial identity (Figure 4C,D and Appendix A). Analysis of signal contributions from specific ligand–receptor pairs showed that while BMP4 signalling predominated from the lens and RPCs, BMP7 signalling was strongest in OSE (Appendix A), which is consistent with the proposed role of BMP7 in eye development and, specifically, in corneal function [49,50]. Meanwhile, TGFβ signalling was strongly and exclusively detected as emanating from the lens and RPCs to target the pE and nOSE cells.

Examination of feature plots showing the expression of *PAX6* and the WNT antagonist *SFRP2* revealed marked overlap in OSE but not nOSE (Figure 4E), and this was similarly represented at a cell-type cluster-specific level (Appendix A), supporting the notion that OSE specification is influenced by SFRP2-dependent WNT inhibition. *BMP4* expression was clearly evident in both the developing epithelial cells and in the lens and retinal precursors (Figure 4E and Appendix A). RA signalling was closely associated with lens and RPCs and OSE, and RA communication was inferred to, but not from, nOSE (Appendix A). Notably, there were significant contributions from multiple different members of the retinoic acid network, consisting of RA-ALDHA1 signalling via various combinations of retinoic acid receptors (RARs) and the cellular retinoic acid binding protein CRABP2 (Appendix A), indicating that RA-induced transcriptional regulation is an important event during the specification of optic vesicle and epithelial cellular zones during the early stages of SEAM development. This suggests a wider role for RA in early SEAM development and is reflective of the known roles of RA signalling in neurogenesis and stem cell biology [51,52].

### 3.5. Transcriptional Regulators and Their Networks Define Early Ocular Cell Identity

Numerous key transcriptional regulators work in conjunction with signalling mediators to influence and underlie early cell fate decisions, and the interplay of these pathways and their associated downstream targets is critical in determining cellular outcomes and the precise orchestration of developmental processes. To investigate the cooperative roles of transcriptional regulators and signalling mediators during the early stages of SEAM formation, we employed SCENIC (single-cell regulatory network inference and clustering), a computational tool which aids in identifying transcription factors (TFs) and their potential target genes (the TF regulon) [18], and sought to explore how key transcriptional regulators that drive cell identity might intersect with mediators of cellular communication. We first aggregated the data across all three timepoints, normalised the individual models to account for variability in sequencing depth, and re-clustered the cells (Figure 5A). We then used SCENIC to infer co-expression modules between TFs and candidate target genes, score the activity of each regulon in each cell, and calculate regulon specificity scores (RSS) [18,53]. Figure 5B shows the top-scoring TFs and their projected regulons, which are specific for the cell clusters annotated in Figure 5A. Cells in cluster 9, a highly pluripotent population, scored strongly for NANOG_(20 g), indicating specific activity of this TF driver and its 20 target genes. These targets include the nodal co-receptor *TDGF1*; *SEPHS1*, which regulates RA signalling; and *SPRY4*, a recently studied ERK-dependent regulator of FGF signalling which is vital for preserving stem cell identity [54,55] (Appendix A).

Cells in cluster 2, which represent +10 day cells that are in the process of differentiation towards a neuroectodermal fate, scored most highly for the LHX5_(22 g) regulon. The LHX5 regulon contains 22 putative targets, including the RA-metabolising enzyme *CYP26A1*, the secreted antagonist *FST* (c.f. Figure 2E,F) and the anteriorly-restricted homeobox repressor *HESX1* (Appendix A). After +1 WK and the establishment of pre-SEAM identity, these newly differentiating neuroectodermal cells (cluster 1) scored most strongly for ETV5_(15 g). *ETV4* and *ETV5* are downstream transcriptional activators of the FGF signalling pathway thought to be required for the timely transition from pluripotency [56], so activation levels here could influence the initiation of neuroectodermal differentiation from pluripotent precursors. Notably, TF-network analyses of TP63-positive SEAM-reporter cells have also identified the ETV5 regulon as specific to early-differentiating neuroectodermal clusters [57]. Later in differentiation, as the neuroectodermal cells adopt more specialised phenotypes, this is then accompanied by a shift in the most highly scoring regulons. After +2 WKs, for example, presumptive retinal cells (cluster 5) scored most highly for VSX2_(16 g), LHX9_(12 g), and VAX2_(15 g), whereas cells tending towards NPC identity (cluster 4) scored most highly for regulons that include EMX2_(41 g) and PRDM16_(15 g) (Figure 5B, Appendix A). Interestingly, in vivo, PRDM16 is thought to cooperate with BMP and WNT signalling cues to regulate neural stem cell behaviour [58].

In the surface ectodermal and early epithelial clusters (6, 7 and 14), constituent cells will give rise to populations of *PAX6*+ ocular and *PAX6*- non-ocular epithelial cells located in Zones 3 and 4 of the mature SEAM. We found that the GRHL1 regulon was strongly correlated with SE/epithelial cells in these clusters, together with the GRHL2, TFAP2B/C, TP63, GATA2/3, OVOL2, and DLX4 regulons (Figure 5B), and we found that several of these regulon sets were highly active in SE cells that expressed *PAX6* (Appendix A). This was particularly pronounced for DLX4_(28 g), where regulon activity strongly correlated with *PAX6* expression in the developing ocular epithelium (Appendix A). We also used our SCENIC output to explore individual genes within selected regulons and found multiple candidates potentially relevant to SEAM development. For example, *MAL2*, a recently described corneal epithelial lineage marker [59], featured in both the GRHL2 and OVOL2 regulons, while *HAPLN1*, which encodes a structural protein that links hyaluronic acid and proteoglycans, featured in the OVOL2, DLX4, GATA3, GRHL1, and GRHL2 regulons, and expression of these factors was restricted to developing epithelial cells (Appendix A). Meanwhile, *MSX2*, a BMP-inducible homeodomain transcription factor involved in anterior segment development of the eye [60], was returned as both a TF driver and regulon target in epithelial populations, with predicted effectors including *GATA2* and *GATA3*, the WNT ligand secretion mediator *WLS*, and *BAMBI*, which encodes a pseudo-receptor for the TGFβR1 family (Appendix A). The smallest cluster in our dataset, cluster 18, was closely associated with developing retinal populations (Figure 5A) and was marked by increased MSX1_(23 g), PAX3_(23 g), PAX7_(11 g), and ZIC1_(15 g) activity (Figure 5B). These factors have traditionally been associated with the developing neural crest [61], but recent investigations into ciliary marginal zone (CMZ) niches in mammalian retinas have also correlated the expression of *MSX1* and *ZIC1* with the presence of a transient CMZ cell population, which is distinct from classical RPCs [62,63], so this too might be reflected in our data.

Finally, to explore the functional relevance of these regulons to the SEAM model in more detail, we performed Metascape analysis [64] on selected gene lists returned from SCENIC analysis. Enriched ontology clusters for GRHL1_(87 g), for example, showed strong correlation with processes encompassing tissue morphogenesis, sensory organ development, neural crest formation, and epithelial cell differentiation (Figure 5C, Appendix A). Notably, there were specific, highly connected nodes representing eye and visual system development, and multiple nodes pertaining to early epithelial differentiation and embryonic growth. We then applied this analysis to a comprehensive list of transcriptional drivers that operate during SEAM development by modifying our input to contain the top specific regulons for all clusters as specified by the RSS (Figure 5B). Representative nodes relating to embryonic head and sensory organ development, cell proliferation and differentiation, gastrulation, and pattern specification were all returned (Figure 5D, Appendix A), alongside individual nodes representing camera-type eye formation and morphogenesis, highlighting the inherent suitability of immature hiPSC-derived SEAMs to model the very earliest stages of eye development.

## 4. Discussion

The intersection of developmentally conserved signal transduction mechanisms and the expression of key transcriptional regulators is key to maintaining the unique characteristics of stem cells. For example, Activin and FGF2 signalling networks play pivotal roles in modulating SMAD2/3 and MAPK pathways to sustain stem cells in an undifferentiated state, while transcription factors such as POU5F1, SOX2, and NANOG are principal regulators of self-renewal [65,66]. In contrast, BMP and WNT signalling pathways, in conjunction with FGF, RA, and TGFβ pathways, are essential for initiating differentiation and guiding lineage commitment. The dynamic interactions between these pathways and their downstream effectors are critical for determining the fate decisions of newly differentiating hiPSC/SEAMs and the generation of ocular cell phenotypes. Our hiPSCs were cultured in maintenance medium for 10 days before this was substituted with differentiation medium to promote initial SEAM formation. However, even prior to the differentiative media switch, there was evidence of transcriptional variation, and there were marked differences in predicted cellular potencies. Cells displaying high stemness scores (e.g., elevated expression of *NANOG*) signalled strongly through Activin, while FGF2 signalling was widespread in all but the most differentiated cells, which instead signalled through FGF8. Meanwhile, RA signalling emanated from a single cluster of FGF8-signalling cells which were transitioning towards neuroectoderm, and this signal was received by cells in all clusters. In this regard, it is interesting to note that diffusible RA signalling can directly repress *FGF8* transcription during development, and this restriction controls axis extension and stimulates neurogenesis in undifferentiated precursor cells [67,68]. Expression patterns in our data suggest that RA might play a similar regulatory role in differentiating hiPSC/SEAMs, with the early neuroectodermal subpopulation acting as a source of RA signalling.

The expression pattern of the Activin antagonist *FST* in the newly differentiating cells is also particularly interesting. Best known for its neural-inducing activity via Spemann’s Organiser [69,70], studies in the chick have recently shown that *Pax6* interacts with *Fst* and *Tgfb2* to form a self-organising ‘Turing’ network which functions to polarise the developing optic vesicle [71,72]. In this model, *Pax6* exerts its master control by directing the expression of *Fst* and *Tgfb2*, and the encoded morphogens reciprocally modulate *Pax6* activity. This network activity has been proposed to underscore not only the ability of *PAX6* to induce ectopic eye development, but also the self-organisation of optic cups from stem cell aggregates in vitro [72]. In our hiPSC/SEAM model, *FST* is robustly expressed in all but the most stem-like of cells. At this stage in the differentiation protocol (i.e., 10 days), *PAX6* expression is far more restricted and is only evident in those cells that have begun to commit to neuroectodermal fates, but it is intriguing to consider how similar networks might contribute to progressive SEAM formation. Recent studies have also begun to explore the evolutionary conservation of human organisers using embryonic stem cell models, and synthetic organiser cells have been engineered to steer mouse ES cell development by incorporating spatially defined morphogen gradients [73,74]. Strikingly, organisers programmed to produce opposing signals from WNT3A or its antagonist DKK1 direct distinct lineage outcomes, whereas a single DKK1 node directs only head-like neuroectodermal fates [74]. Consistent with this, we find a cluster of *DKK1*-expressing cells in our +10-day hiPSC/SEAMs that are tending towards neuroectodermal differentiation.

Early SEAM differentiation autonomously results in predominant differentiation toward neural fates, with 85% of cells at +1 WK and 72% of cells at +2 WKs displaying neuroectodermal characteristics. Indeed, even during the hiPSC maintenance phase of culture, there was evidence of differentiation toward presumptive neuroectodermal fates. Analysis of the communication networks that operate over developmental time reveals the signalling mechanisms that underlie this. The widely accepted (but often debated) ‘default model’ for neural induction informs us that in the absence of instruction, ectodermal cells will adopt a neural fate. Consistently central to the arguments relating to neural induction is the role of BMP inhibition. Conversely, BMP signalling is required for the promotion of non-neural ectodermal fates, and there is significant interplay with other signalling mediators, including WNTs and FGFs [75,76]. While a tendency towards neural identity is indeed evident, after 1 week of differentiative culture, we are able to characterise populations of surface ectoderm and differentiating epithelial cells in our data, and a proportion of these cells express *PAX6*, indicating that they are adopting ocular fates in the absence of exogenous stimulation. SEAMs kept over longer periods in culture have been used to generate functional corneal epithelium [1,2,4], and while this does require a later change in the culture medium to promote continued epithelial differentiation, our data from early SEAMs indicates that the process of epithelial lineage commitment is ongoing prior to this. Analysis of ocular surface ectodermal-like cell morphology in both the immature pre-SEAM and the mature SEAM has indicated that if SE-derived epithelial cells fail to appear in the pre-SEAM by day 10 of differentiation, they are then entirely absent from Zone 3 of the mature SEAM after six weeks [37]. This observation suggests that intra- and inter-zonal communication is a strict requirement for proper SEAM initiation, and that, in the absence of exogenous factors, SE cells must differentiate from a common progenitor pool as a result of instructive signalling and/or co-transcriptional regulatory events. Our communication analyses here support that this is likely a result of directed BMP signalling together with the inhibition of WNT in ocular surface epithelium, which is accompanied by overlapping expression patterns of *PAX6* and *SFRP2*.

Integrated analysis of SCENIC transcription factor–regulon relationships indicates that the grainyhead-like transcription factors GRHL1 and GRHL2 also play important roles in epithelial specification in the early SEAM, along with DLX4, GATA2/3, TFAP2A/C, and OVOL2. As embryonic stem cells lose pluripotency and differentiate into committed surface ectodermal cells, chromatin accessibility and transcriptional activity increase for AP2 factors (*TFAP2A/C*), GATA factors (*GATA2/3*), and GRHL factors (*GRHL1/2*), while *POU5F1*, *NANOG*, and *SOX2* lose their influence [77]. Notably, several AP2, GATA, and GRHL factors have also been shown to be BMP-inducible [78,79,80]. OVOL2 acts downstream of BMP signalling and is part of a core transcriptional network implicated in the direct reprogramming of human fibroblasts into corneal epithelial cells, and also contributes to a finely balanced corneal GRHL2-OVOL2-ZEB1 feedback circuit which is closely associated with proper WNT signalling [81,82], so it is conceivable that similar circuits operate during the establishment and maintenance of ocular SEAMs in culture. Notably, in our data, *OVOL2* also features in the SCENIC regulon for GRHL2, and *PAX6* is expressed by a subset of epithelial cells which are active for the OVOL2 and GRHL1/2 regulons, but show less well-defined *OVOL2*/*GRHL1/2* gene expression profiles. This highlights one of the many advantages of a combinatorial approach to single-cell analysis, given that wider regulon specificity does not necessarily always directly correlate with simple gene expression when considered in isolation. DLX4_(28 g) regulon activity was also particularly pronounced in ocular epithelial cells, and *DLX4* has previously been shown to be downregulated in *PAX6*-knockdown limbal epithelial stem cells, indicating that it operates downstream of *PAX6* [83,84]. Our analyses here suggest that *DLX4* could be a useful marker for OSE characterisation. The incorporation of TF-to-gene analysis can also facilitate the identification of functionally relevant cell populations that might otherwise be overlooked by standalone expression analyses. For example, our SCENIC results indicate that a small population of cells within the larger neuroectodermal supercluster show elevated specificity scores for factors with potential links to CMZ development, such as MSX1 and ZIC1. In our previous study, we speculated that a CMZ-like zone might spontaneously generate during SEAM maturation, and in UMAP renderings of mature assemblies this locates to the border between retinal pigment epithelium and neural retina [5]. Comparable CMZ-like populations have since been independently reported in hiPSC-derived SEAM assemblies after 14 days of culture [85]. Our results here support the finding that precursors to this population could emerge during the early stages of SEAM culture, so in future work, it would be interesting to ascertain how these cells originate and mature over developmental time. This is particularly relevant given the recent discovery that retinal stem cells capable of regeneration are located in CMZ-like niches in foetal retinas and in retinal organoids [86].

Single-cell transcriptomic analysis of growing SEAMs has revealed striking correlations with several facets of ocular development, such as the sequential delineation of neural retina and retinal pigment epithelium and temporally conserved emergence of retinal ganglion cells and cone photoreceptors [5]. Our work here extends these findings and suggests that there is similar conservation of early cellular communication in vitro. This is clearly illustrated in, for example, 2-week-old SEAMs, where analyses of communication patterns between secreting and target cells reveal a shift in cell type-specific BMP signalling, with BMP4 signal predominating from cells in the mixed lens and retinal progenitor group rather than the earlier surface epithelial cells. During early establishment of the lens, multipotent precursors begin to give rise to distinct populations, and these cells are guided to form the lens placode through a regulatory network controlled by *PAX6*. This formation is influenced by paracrine signals from the prospective retina, autocrine BMP signals, RA signalling, and ocular suppression of WNT signalling within the presumptive lens ectoderm [46,48]. Our data show that these patterns are strikingly well conserved in developing SEAMs. We also observed elevated canonical WNT signalling in non-ocular ectoderm, and this is, in vivo, stimulated by migrating neural crest cells [87], which are present in limited numbers at this stage of SEAM formation. Periocular neural crest-derived TGFβ signalling is also thought to suppress *PAX6* to align the lens and optic vesicle during embryogenesis [87], and in our data, we found that incoming TGFβ signalling was restricted to proliferating epithelial cells and non-ocular SE cells, which typically do not actively express *PAX6*. Inferred WNT signalling was strongest from residual pluripotent cells, targeting proliferating epithelium, nOSE, and OSE in order of decreasing strength, and analysis of differentiation dynamics in early differentiating SEAMs has indicated a developmental trajectory that flows from proliferating SE through to nOSE and thereafter to OSE [57], so it is particularly interesting to consider how these communication patterns define the SEAM as it self-organises and matures in vitro. RA signalling patterns are also of considerable interest in our data. While RA has been well studied in the context of head and eye development, the contribution of specific receptor engagement to RA-induced transcriptional regulation is not yet understood in detail. However, recent work modelling the global sensitivity of these receptors suggests that this may depend on the intracellular concentration of RA, which itself is dependent on cell type [88]. In our study, after 2 weeks of culture, there was pronounced multi-engagement RA signalling inferred from cells in retinal and OSE clusters. In vivo, secreted RA produced in the retina by ALDH1A1 is thought to act on cells in the periocular mesenchyme to regulate the expression of genes controlling corneal morphogenesis during anterior eye formation [89,90], and periocular mesenchymal cells transcriptionally cluster with developing ocular epithelium in early SEAMs [5], indicating that this too might be mirrored by our results. We also note that after 2 weeks of culture, RA signalling was detected to, but not from, nOSE. This is presumably because the expression of *ALDH1A1* is restricted to the *PAX6*+ presumptive ocular cells of the developing SEAM. In early-stage SEAM differentiation protocols, WNT activation via CHIR, when combined with exogenous RA, results in an increase in the number of P63+ epithelial cells in the absence of PAX6+ ocular cells, while a combination of WNT inhibition and exogenous BMP4 initiates OSE formation [37], which reconciles with the cellular communication data presented here. Functional perturbation data from human organoid models also directly support inferred signalling hierarchies relevant to eye development. Early modulation of developmental pathways has been shown to have profound effects on cellular composition and cell-class abundance in hESC-derived organoids, for example, with staggered activation of BMP followed by WNT producing a significantly higher proportion of retinal cells compared to BMP activation alone [91]. Meanwhile, the use of small-molecule inhibitors targeting TGFβ and WNT signalling, in combination with FGF activation, can promote the differentiation of corneal epithelial cells from hiPSCs [92]. Indeed, pioneering protocols originally developed to create self-organising optic cups relied on in vivo experimental knowledge and the manipulation of various signalling pathways [93,94,95], and the adoption of multi-omic profiling now facilitates detailed exploration of these pathways and the cellular and molecular similarities that exist between in vitro models and the developing eye.

Conventional SEAMs are grown on a substratum composed of heterotrimeric laminin α5β1γ1 (LN511E8) subunits. However, the use of different laminin isoforms can influence cell phenotypes in ocular SEAMs. For example, LN211E8 promotes differentiation into neural crest via activation of WNT, whereas hiPSCs grown on LN332E8 preferentially form epithelium [38,39]. This propensity of cells to grow on alternative isoforms can be correlated with integrin expression profiles, and moreover, with laminin subunit expression profiles in vivo [39]. By analysing cell and ECM communication probabilities in transcriptomically resolved clusters, we are able to extract information that could help further refine the specific parameters used in the directed differentiation of ocular populations. For example, we observed no signalling via LAMA2 in our data, and this may partly explain why we have been unable to retain significant numbers of periocular neural crest-derived corneal endothelial cells using standard SEAM culture conditions. HSPG signalling was particularly active in cells belonging to the surface ectodermal clusters, and HSPGs are required for the proper establishment of specific morphogen gradients in development, including in the eye [96,97]. Recent studies have suggested that proteoglycan/glycosaminoglycan signalling might potentiate the formation of corneal epithelial cells from hiPSC/SEAMs [98,99], so here, too, there is an opportunity to interrogate signalling modules in order to refine SEAM culture. Beyond providing mechanistic insight into early ocular lineage specification, the signalling hierarchies and regulatory programmes inferred here also have practical relevance for both disease modelling and translational applications. First, defining when and where pathways such as BMP, WNT, RA, and TGFβ predominate during SEAM self-organisation can help rationalise and refine directed differentiation strategies by informing the timing, direction, and combinations of pathway modulation used to bias fate choice and promote maturation. Second, because many ocular disorders arise from disrupted early patterning and aberrant tissue interactions, the source-target signalling relationships resolved in this study provide a framework for modelling developmental defects in vitro and for testing how genetic perturbations could alter early communication states. Indeed, SEAM cultures have recently been employed to map the expression of disease-associated markers for a broad range of ocular disorders, including corneal dystrophy, anterior segment dysgenesis, cataract, and inherited retinal diseases/optic atrophy [85]. Finally, given the demonstrated translational potential of SEAM-derived ocular epithelial cells [4], a clearer understanding of the signalling environment that supports correct specification and maturation may help improve the robustness, consistency, and functional quality of therapeutic products intended for regenerative use.

Our analyses here also provide a resource to interpret the potential functions of less well-characterised signalling mediators and/or their interactors. For example, FLRT signalling was consistently highlighted by our communication analyses, and although FLRTs have been described in the eye, how they operate in this context, either by homophilic adhesion or heterotypic ligand binding, is not well understood. While the involvement of pathways such as BMP, WNT, FGF, and RA in ocular development is well established, by leveraging single-cell transcriptomics in conjunction with cell–cell communication inference, our study provides new insights into the spatiotemporal dynamics, source–target specificity, and developmental context of these signals within a human in vitro model, whilst highlighting the potential involvement of lesser studied mediators. In considering how SEAM development aligns with human embryogenesis, we note that direct equivalence between in vitro and in vivo timelines is inherently approximate. However, based on early transcription factor expression patterns, the separation of neuroectodermal and non-neural ectodermal cell identities, and the emergence of defined lineage markers, we speculate that SEAMs at +1 to +2 weeks of differentiation broadly correspond to ~3–4 post-conception weeks in vivo. This developmental window encompasses eye field specification, optic vesicle formation, and the onset of lens induction, and precedes the segregation of distinct neural retinal (*VSX2*) and RPE (*MITF*) territories, which occurs from around 5 pcw [95] or after 4 weeks of SEAM cell culture [5]. Previous work modelling *PAX2* and *PAX6* distribution patterns has indicated that SEAMs at 3–4 weeks correspond to ~4–5 weeks in vivo [57], consistent with our observations. *PAX2*, which marks the ventral optic cup and later becomes restricted to the optic fissure and stalk [100], showed no defined expression in our early SEAMs, supporting the interpretation that these cultures model pre-cup stages of ocular development.

There are some limitations to our study. First, our analyses were conducted using a single hiPSC line (201B7), which has been extensively validated for ocular differentiation but does not allow for the assessment of inter-line variability. Second, although the use of stringent quality control metrics and SCTransform-based normalisation across multiple time points helps mitigate downstream technical artefacts while preserving biological heterogeneity, tools such as SCENIC and CellChat can remain sensitive to data sparsity and noise. As with all transcriptome-based analyses, our study also reflects steady-state mRNA levels and cannot directly account for post-transcriptional regulation, isoform-specific expression, or protein activity. Furthermore, to better understand the interplay between gene activity, cell–cell signalling and tissue architecture, future studies will require the integration of spatially resolved transcriptomic approaches that preserve positional context. Our analyses here also do not incorporate direct perturbation data, so future work could apply temporal pathway inhibition or ligand supplementation to explore specific source–target interactions predicted by our communication networks. Finally, while hiPSC/SEAMs recapitulate many aspects of early human eye development, they remain an in vitro model that cannot fully mirror the complexity of native tissue organisation and morphogenesis in the developing eye. Nevertheless, the use of trajectory-aware, high-resolution single-cell data provides key insights into conserved signalling and transcriptional mechanisms that underpin ocular lineage specification.

## Figures and Tables

**Figure 1 cells-15-00104-f001:**
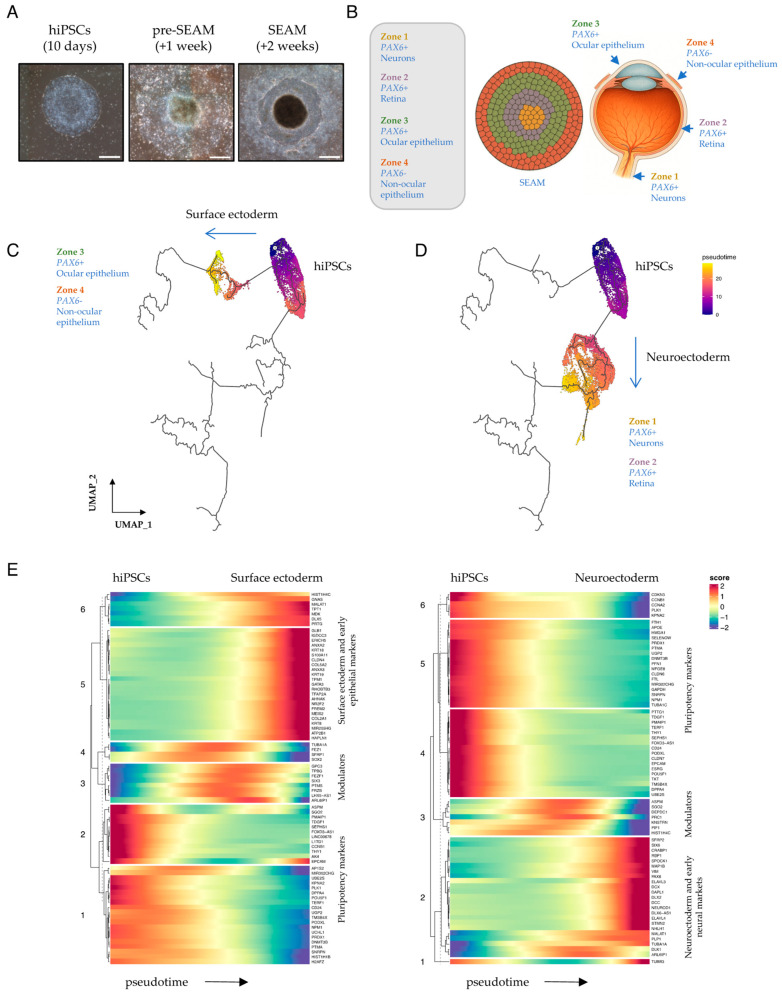
**Surface ectodermal and neuroectodermal specification in early SEAMs.** (**A**) Representative phase-contrast images, modified from [5], showing typical differentiating hiPSCs/SEAMs after 10 days, +1 WK or +2 WKs of culture. Scale bar: 500 μm. (**B**) Schematic depicting the relationship of a 4-zone mature SEAM to cells of the eye. (**C**,**D**) Monocle3-constructed developmental pseudotime trajectories. Cells were scored with respect to an inferred trajectory through to 12-week SEAM maturity by setting clear_cds = F. The root of the trajectory is labelled [1]. (**E**) Heatmaps showing markers that change as a function of Monocle3 pseudotime along each independent lineage branch. Pseudotime direction is depicted by arrows and heatmap scores indicate relative expression over time.

**Figure 2 cells-15-00104-f002:**
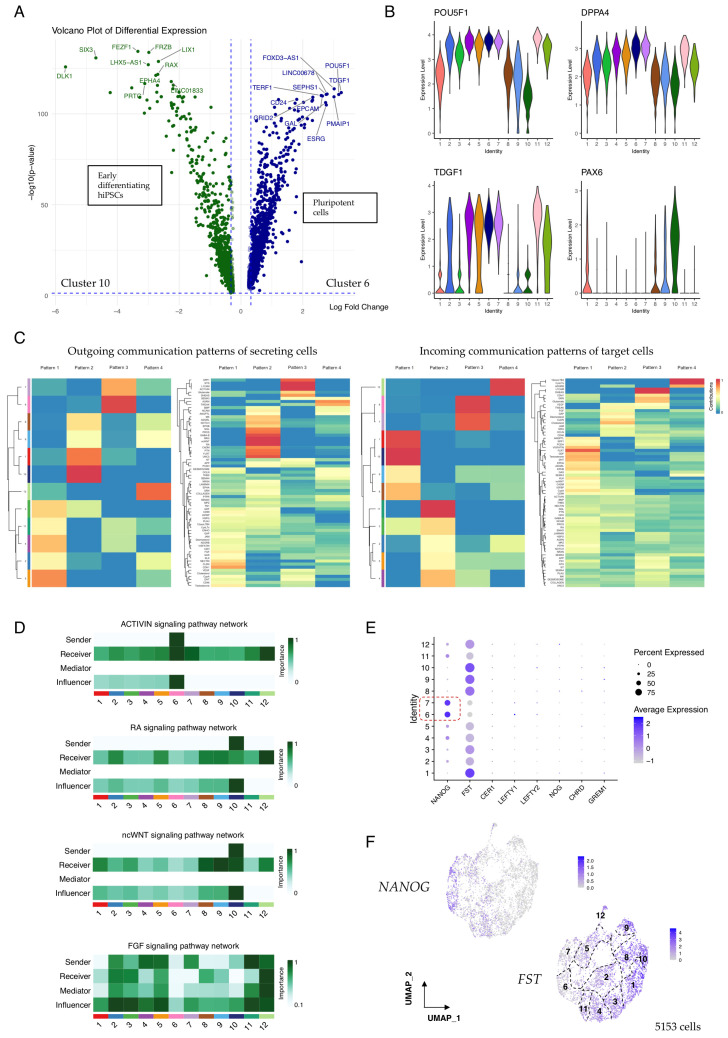
**Communication pattern analysis in 10-day-old hiPSC/SEAMs.** (**A**) Volcano plot showing the top differentially expressed genes in cells of cluster 6 (highly pluripotent) vs. cluster 10 (early differentiating). (**B**) Violin plots showing the expression of key pluripotency markers and reciprocal expression of *PAX6*. (**C**) CellChat-constructed heatmaps showing cell and communication patterns of secreting and target cells. Colours represent relative pattern contribution scores, scaled 0–1. (**D**) Heatmaps showing contribution scores of selected pathways, calculated using netAnalysis_computeCentrality. (**E**) Dot plot showing the expression of pathway antagonists according to Seurat annotation. Dashed box indicates highly pluripotent cell clusters. (**F**) Feature plots showing the expression of *NANOG* and *FST*. Seurat clusters are numbered.

**Figure 3 cells-15-00104-f003:**
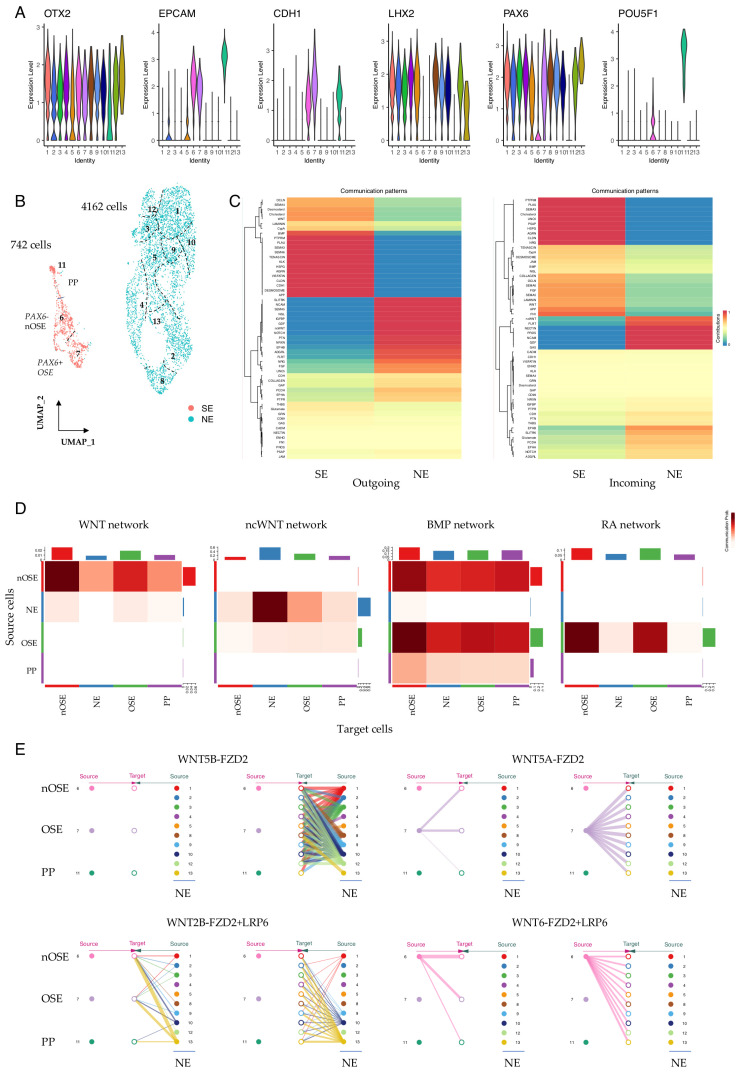
**Communication pattern analysis in +1 WK pre-SEAMs.** (**A**) Violin plots showing expression of neuroectodermal (*OTX2*, *LHX2*), surface ectodermal (*EPCAM*, *CDH1*), and pluripotency (*POU5F1*) markers in +1 WK SEAMs. (**B**) UMAP representation of the contribution of cells to SE and NE lineages. Seurat clusters are numbered. (**C**) CellChat-constructed heatmaps showing incoming and outgoing communication between SE and NE cells. Colours represent relative pattern contribution scores, scaled 0-1. (**D**) Communication probabilities for selected pathways. Colour intensity indicates inferred communication probability between source and target cell groups. (**E**) Hierarchy plots showing signalling contribution of WNT5A and WNT5B (non-canonical) and WNT2B and WNT6 (canonical). Source cells are shown at the outer edges of each plot, with target cells in the centre. PP; pluripotent; SE, surface ectoderm; NE, neuroectoderm; OSE, ocular surface ectoderm; nOSE, non-ocular surface ectoderm.

**Figure 4 cells-15-00104-f004:**
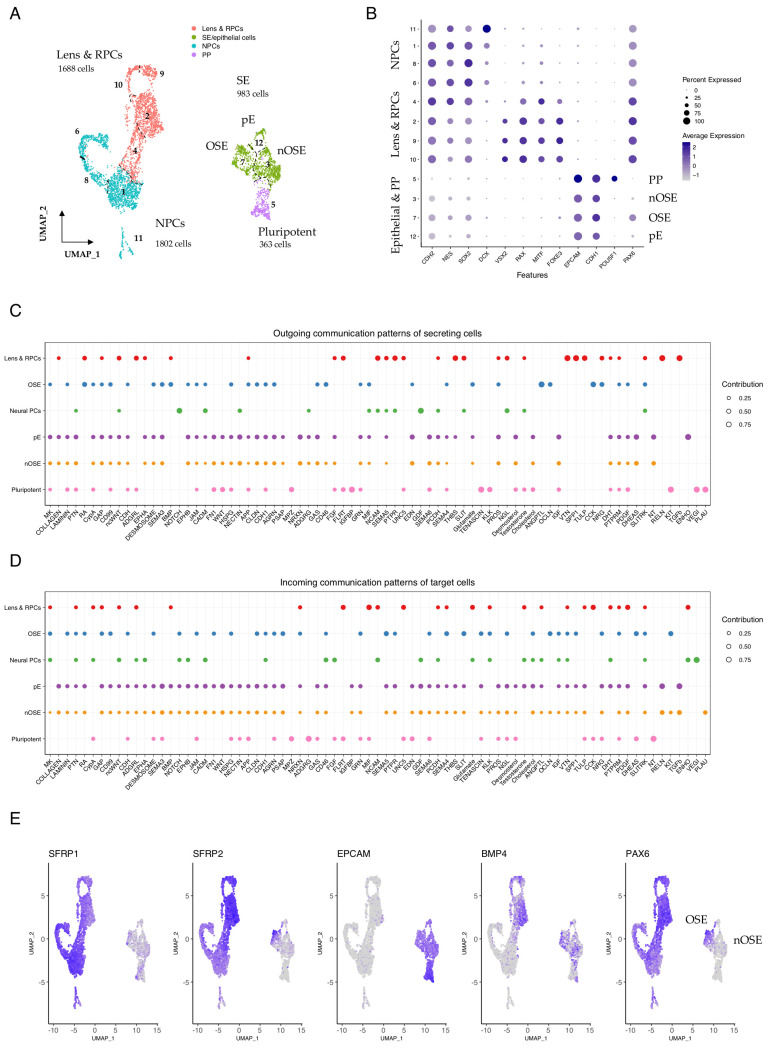
**Communication patterns in +2 WK SEAMs.** (**A**) UMAP representation of cells in +2 WK SEAMs, annotated according to cell type. Seurat clusters are numbered. (**B**) Dot plot showing average expression of markers in epithelial, pluripotent, lens and retinal progenitor cell, and neural progenitor cell subgroups. (**C**) Dot plot illustrating global outgoing communication patterns. Dot size is indicative of contribution score. (**D**) Dot plot illustrating global incoming communication patterns. Dot size is indicative of contribution score. (**E**) Feature plots showing overlapping expression of *SFRP2* and *PAX6* in developing ocular SE and *BMP4* in developing lens and RPCs. PP, pluripotent; NE, neuroectoderm; SE, surface ectoderm; OSE, ocular surface ectoderm; nOSE, non-ocular surface ectoderm; pE, proliferating epithelium; RPC, retinal progenitor cells; NPC, neural progenitor cells.

**Figure 5 cells-15-00104-f005:**
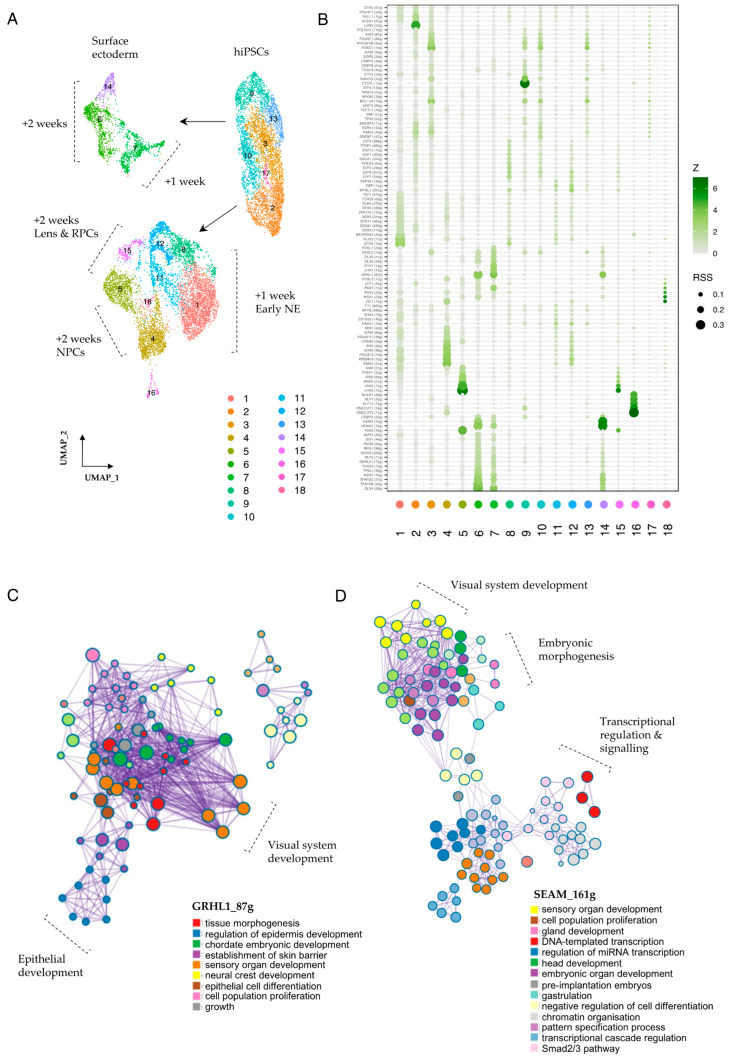
**Transcriptional regulators in early SEAM development.** (**A**) UMAP representation of re-clustered data (+10 days, +1 WK, +2 WK). Seurat clusters are numbered. (**B**) Regulon specificity score (RSS) plot generated using SCENIC output, indicating regulons specific to numbered Seurat clusters. RSS and Z-scores are indicated, and only ‘high-confidence’ annotations are plotted. (**C**) Metascape analysis showing network of process-enriched terms. Relevant nodes are labelled. Input list = genes contained in the GRHL1_(87 g) regulon. (**D**) Metascape analysis showing network of process-enriched terms. Relevant nodes are labelled. Input list = all TF drivers shown in RSS plot (**B**).

## Data Availability

scRNA-seq datasets have been deposited in NCBI GEO under accession number GSE263987 (WK0, WK1, and WK2 represent +10 day, +1 WK, +2 WK datapoints).

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
