# Peer review of "Single-Cell RNA-Seq Reveals Conserved Cellular Communication Mechanisms Governing Ocular Lineage Specification from Human iPS Cells"

_cells, 2026, doi:10.3390/cells15020104_

Round 1
Reviewer 1 Report
Comments and Suggestions for Authors
Howard et al.'s current study expands upon their earlier work published in Nature Communications Biology (2024). This study offers a deeper understanding by analyzing cell-cell communication and transcriptional regulators that guide the development of ocular lineages from human PSCs. By integrating CellChat and SCENIC analyses across the differentiation timepoints, the authors uncover how developmental pathways coordinate with transcription factors to guide ocular lineage specification. I found this work interesting because it offers mechanistic insights into how stem cells and their differentiated progenitors regulate, signals and communicate during early eye development. The experimental evidences very clear and the manuscript is well written. Please incorporate the following suggestions into the current version of the manuscript.
- Methods: Verify these matching catalogue numbers. 0.5 mM EDTA/PBS (13567-84, Nacalai Tesque) and TrypLE Express (13567-84, Nacalai Tesque)
-
Line 173. PRC1 is not a direct antagonist of RA. As the authors highlighted, PRC1 is an epigenetic repressor, but it down-regulates transcription by repressing RA-target genes.
-
hiPSC; iPSC/SEAMs terms are used inconsistently; make this uniform throughout.
-
The authors report using single-cell data from one hiPSC line (201B7). Does this raise the possibility of batch effects due to a lack of replicate validation?
Are the SCENIC/CellChat sensitive to batch effects? To discuss. -
Describe the limitations of the study (post-translation modification/transcripts not translated; isoform ambiguity, comparison of cellular models/human retina etc)
Author Response
Methods: Verify these matching catalogue numbers. 0.5 mM EDTA/PBS (13567-84, Nacalai Tesque) and TrypLE Express (13567-84, Nacalai Tesque)
We apologise for the error and have corrected this in the text.
Line 173. PRC1 is not a direct antagonist of RA. As the authors highlighted, PRC1 is an epigenetic repressor, but it down-regulates transcription by repressing RA-target genes.
We thank the reviewer for this important clarification and agree that PRC1 is not a direct antagonist of RA signalling, and instead is an epigenetic repressor which acts to repress transcription of RA-responsive genes. We have modified the text to reflect this (lines 173-174)
hiPSC; iPSC/SEAMs terms are used inconsistently; make this uniform throughout.
We have modified the manuscript for consistency. Specifically, we use ‘hiPSC/SEAMs’ where we are directly referring to the earliest (+10 day) timepoint, as this stage is primarily concerned with pluripotent cells, and ‘SEAMs’ thereafter.
The authors report using single-cell data from one hiPSC line (201B7). Does this raise the possibility of batch effects due to a lack of replicate validation? Are the SCENIC/CellChat sensitive to batch effects? To discuss.
We thank the reviewer for raising this important point. We acknowledge the limitation of using a single hiPSC line (201B7); however, this line is extensively characterised, routinely used in SEAM-based ocular differentiation protocols, and has been applied in previous clinical studies. To minimise technical variation, we applied stringent quality control, SCTransform-based normalisation, and tools such as PrepSCTFindMarkers to account for variations in sequencing depth across time points. This workflow is designed to preserve biological heterogeneity while mitigating technical effects. We note that standard batch correction tools are mainly designed to integrate comparable cell states across biological or technical replicates. In our study, each time point represents a biologically distinct developmental stage, so applying batch correction across time points could remove genuine temporal signal. Additionally, while SCENIC and CellChat analyses can be sensitive to sparse or noisy data, we used conservative parameters and interpreted findings within biologically consistent pseudotemporal trajectories. We have added a discussion of these limitations in the revised manuscript (lines 839–845) and agree that future studies incorporating multiple hiPSC lines would be valuable to confirm and extend these findings.
Describe the limitations of the study (post-translation modification/transcripts not translated; isoform ambiguity, comparison of cellular models/human retina etc)
We thank the reviewer for this helpful suggestion. In the revised manuscript, we have added a paragraph in the discussion (lines 845–857) highlighting the key limitations of our study. These include the inability of our transcriptome-based analyses to capture post-transcriptional regulation, isoform-specific expression, or protein activity, as well as the limitations inherent to in vitro models compared with native human tissue. We also note the potential of spatial transcriptomics to resolve gene activity and intercellular communication in the context of positional information.
Reviewer 2 Report
Comments and Suggestions for Authors
This manuscript explores early cell fate decisions in eye development by using human induced pluripotent stem cell (hiPSC)-derived eye-like organoids, analyzed through transcriptomic and single-cell approaches. The topic is both timely and relevant, as stem-cell-based organoid models offer a unique window into human developmental processes that are otherwise difficult to study. The work has the potential to make meaningful contributions to developmental biology, regenerative medicine, and ocular disease research. With that in mind, here are several observations and recommendations:
The combination of hiPSC-derived eye organoids with single-cell analyses is a powerful approach and builds on a field that is still rapidly evolving. However, the manuscript could benefit from more clearly highlighting what sets this study apart from previous work. For example, it would be helpful to clarify whether the signaling mediators identified—such as Activin, FGF, BMP, WNT, and retinoic acid—provide new mechanistic insights or largely confirm what has already been observed.
The focus of the study appears to be on early differentiation events. The manuscript would be strengthened by specifying the developmental window being modeled (for instance, which embryonic stage the organoids correspond to) and providing some context on how well these organoids mimic the in vivo human optic vesicle or retinal progenitor states.
Author Response
The combination of hiPSC-derived eye organoids with single-cell analyses is a powerful approach and builds on a field that is still rapidly evolving. However, the manuscript could benefit from more clearly highlighting what sets this study apart from previous work. For example, it would be helpful to clarify whether the signaling mediators identified—such as Activin, FGF, BMP, WNT, and retinoic acid—provide new mechanistic insights or largely confirm what has already been observed.
We thank the reviewer for this helpful observation. While many of the signalling pathways identified by our analyses are well established as core regulators of ocular development, our goal was to evaluate how faithfully these pathways are recapitulated within the SEAM model, thereby highlighting its utility for studying early human eye development. However, we agree that this distinction could be made more explicit. To address this, we have now clarified in the revised discussion that our framework allows for additional insight into the spatiotemporal dynamics, source/target specificity, and context-dependent activity of these conserved pathways. Moreover, this approach reveals additional, underexplored signalling mediators and regulatory interactions. This clarification has been added to the revised manuscript (lines 817–823).
The focus of the study appears to be on early differentiation events. The manuscript would be strengthened by specifying the developmental window being modeled (for instance, which embryonic stage the organoids correspond to) and providing some context on how well these organoids mimic the in vivo human optic vesicle or retinal progenitor states.
We agree that aligning the SEAM differentiation timeline with human embryonic development is important for contextualising our findings. While direct equivalence between in vitro time points and in vivo stages is inherently approximate, particularly in a two-dimensional model that only partially mimics whole eye development, previously reported expression patterns of PAX2 and PAX6 in SEAMs after 3-4 weeks of culture suggest that this timepoint corresponds to 4-5 pcw (Kamuro et al., Comms Biol, 2025). Based on marker gene expression profiles, morphogen signalling activity, and developmental trajectory analyses, we speculate that SEAMs at +1 to +2 weeks correspond to approximately 3-4 pcw. This window encompasses eye field specification, optic vesicle formation, the onset of lens induction, and the establishment of presumptive ocular versus non-ocular surface ectodermal fates. The co-expression of MITF and VSX2 at +WK2, for example, is consistent with the optic vesicle stage prior to clear compartmentalisation of neural retinal vs. RPE territories, which occurs from around 5 pcw in human eye development, and after +4WKs in SEAMs. We now discuss this in the revised manuscript (lines 823-837).
Round 2
Reviewer 2 Report
Comments and Suggestions for Authors
This manuscript presents a thorough and technically strong single-cell transcriptomic study of early ocular lineage specification using the human iPSC-derived SEAM model. By combining trajectory analysis (Monocle3), cell–cell communication profiling (CellChat), and transcriptional regulatory network analysis (SCENIC), the authors build a clear and convincing framework for how key, conserved signaling pathways—such as Activin, FGF, BMP, WNT, and retinoic acid—work together to guide early cell fate decisions.
The study is thoughtfully designed and carefully executed, and the results are biologically coherent and well supported by the data. Importantly, this work provides valuable insight into early human eye development using an accessible and relevant stem-cell–based system. Overall, the manuscript makes a meaningful contribution to the field and is well suited for publication in Cells, pending minor revisions.
-
The authors should clarify whether any functional perturbation data (e.g., BMP, WNT, or RA modulation) already exist in the literature that directly support their inferred signaling hierarchies.
-
If new wet-lab validation is beyond scope, this limitation should be explicitly acknowledged in the discussion. Add a short paragraph discussing how the inferred signaling networks could be experimentally tested (e.g., pathway inhibition or ligand supplementation).
- Given the strong mechanistic insights, the authors could expand on how these findings inform disease modeling, developmental disorders, or therapeutic differentiation protocols (e.g., optimizing corneal epithelial differentiation).
- The discussion briefly mentions relevance to regenerative medicine and transplantation. I recommend adding a short paragraph linking these findings to translational applications.
- Supplementary figures are extensive and valuable; however, adding a summary table of key signaling pathways per time point would help readers navigate the data.
Author Response
The authors should clarify whether any functional perturbation data (e.g., BMP, WNT, or RA modulation) already exist in the literature that directly support their inferred signaling hierarchies.
We thank the reviewer for this suggestion. Functional studies relevant to early ocular differentiation support the involvement of several key pathways highlighted by our inferred signalling hierarchies. For example, previously published data show that in early-stage SEAM differentiation protocols, WNT activation in combination with exogenous RA increases P63-positive epithelial cells in the absence of ocular cells, whereas WNT inhibition combined with exogenous BMP4 initiates ocular surface ectoderm formation (Kobayashi et al., 2002). Similarly, use of small-molecule inhibitors (e.g. WNT, TGFb) has been shown to promote corneal epithelial cell differentiation from hiPSCs. Human organoid studies using single cell sequencing also show that early modulation of conserved developmental pathways can produce predictable shifts in cell-class abundance and lineage composition (Tresenrider et al., 2023). We discuss this in lines 759 to 775 of the newly revised document.
If new wet-lab validation is beyond scope, this limitation should be explicitly acknowledged in the discussion. Add a short paragraph discussing how the inferred signaling networks could be experimentally tested (e.g., pathway inhibition or ligand supplementation).
We agree with the reviewer that experimental validation would strengthen interpretation of inferred signalling interactions, but this is beyond the scope of the current study. We have therefore explicitly acknowledged this limitation in the discussion by noting that our analyses do not incorporate direct perturbation data, and we outline how future studies could apply temporal pathway inhibition or ligand supplementation to test specific source-target interactions predicted by our communication networks (lines 850-852).
Given the strong mechanistic insights, the authors could expand on how these findings inform disease modeling, developmental disorders, or therapeutic differentiation protocols (e.g., optimizing corneal epithelial differentiation).
We thank the reviewer for this helpful comment. We have expanded the discussion to clarify how the inferred signalling hierarchies and regulatory programmes may inform disease modelling and the refinement of therapeutic differentiation protocols. Specifically, we now highlight how defining when and where pathways such as BMP, WNT, RA and TGFβ predominate during SEAM self-organisation can guide the timing and combinations of pathway modulation used to bias fate choice and promote maturation. We also discuss how the SEAM model system has been used to explore gene expression relevant to a broad range of ocular disorders (Replogle et al., 2025). (lines 791 to 807 in newly revised manuscript)
The discussion briefly mentions relevance to regenerative medicine and transplantation. I recommend adding a short paragraph linking these findings to translational applications.
We thank the reviewer for this recommendation. We have added a section linking our findings to translational applications by explaining how improved understanding of early signalling environments and source-target communication states may help improve robustness, consistency, and functional quality of SEAM-derived ocular epithelial products intended for regenerative use (lines 807–811).
Supplementary figures are extensive and valuable; however, adding a summary table of key signaling pathways per time point would help readers navigate the data.
To aid navigation, we have now included a new Supplementary summary table (Table S2) featuring key signalling pathways discussed in the manuscript. Pathway-level CellChat communication matrices were generated at cluster resolution and then collapsed into broader cell-group categories, with relative sender and receiver activity shown as +, ++, +++ (scaled within each pathway). We refer to this table in lines 127-128 of the revised manuscript.